# Conceptualisation and psychometric evaluation of positive psychological outcome measures used in adolescents and young adults living with HIV: a mixed scoping and systematic review protocol

Jermaine M Dambi [1], Frances M Cowan [2,3] Faith Martin [4]
Sharon Sibanda [2], Victoria Simms [5], Nicola Willis [6], Sarah Bernays [7,8]
Webster Mavhu [2,3]

For numbered affiliations see end of article.

**Correspondence to**
Jermaine M Dambi;
jermainedambi@gmail.com

## ABSTRACT

**Introduction** Sub-Saharan Africa bears the greatest burden of HIV. Concomitant mental disorders are common, necessitating the integration of mental healthcare into routine HIV care. Consequently, it is necessary to holistically evaluate the mental health of adolescents and young adults living with HIV (AYALHIV, 10–24 years old) by measuring negative and positive psychological constructs (eg, anxiety and self-acceptance, respectively). There has been a proliferation of positive psychological outcome measures, but the evidence of their psychometric robustness is fragmented. This review, therefore, seeks to (1) identify positive psychological outcomes used in AYALHIV in sub-Saharan Africa and map the constructs onto corresponding measures and (2) critically appraise the psychometrics of the identified outcomes

**Methods and analysis** This mixed review will be done in two parts. First, a scoping review will identify positive psychological outcomes and map them onto corresponding outcome measures. Subsequently, we will systematically evaluate the psychometric properties of the outcomes identified from the scoping review. Independent and blinded reviewers will search articles in PubMed, Scopus, Web of Science, Africa-Wide Information, CINAHL, PsychINFO and Google Scholar from inception through 30 September 2022. Thereafter, separate independent reviewers will screen the retrieved articles. We will apply a narrative synthesis to map the key constructs emerging from the scoping review. For the systematic review, the risk of bias across studies will be evaluated using the COnsensus-based Standards for the selection of health Measurement INstruments (COSMIN) checklist. The quality of the psychometric properties will be rated using the COSMIN checklist and qualitatively synthesised using the modified Grading of Recommendations Assessment, Development and Evaluation checklist.

**Ethics and dissemination** No ethical approvals are needed. The mixed-review outputs will collectively inform the development, implementation and evaluation of bespoke interventions for AYALHIV. Review outcomes will be disseminated in a peer-reviewed journal, on social media and through policy briefs.

## STRENGTHS AND LIMITATIONS OF THIS STUDY

⇒ Application of a systematic methodology guided by standardised guidelines.
⇒ Utilisation of multiple data sources.
⇒ Article screening and data collection will be performed in duplicate and independently.
⇒ Involvement of adolescents and young adults living with HIV in the review and dissemination.
⇒ The exclusion of studies not published in English; this introduces language bias.

**PROSPERO registration** CRD42022325172.

## INTRODUCTION

HIV remains a global health problem, with low-income and middle-income countries disproportionately affected.[1] Furthermore, the burden of HIV in young people in low-resource settings, particularly in the sub-Saharan Africa (SSA) region, remains high.[2–4] Adolescence and early adulthood are challenging developmental stages, with the burden of navigating life challenges often even greater for adolescents and young adults living with HIV (AYALHIV, 10–24 years old).[1 2 5] For instance, AYALHIV face multiple biopsychosocial challenges, including stigma, negotiating reproductive health, socio-economic deprivation, violence and other difficulties.[1 2 6] Also, with the increased availability and uptake of highly effective antiretroviral therapy, HIV has evolved into a long-term condition with a concomitant surge in comorbid non-communicable diseases.[5 7] For example, common mental disorders, including anxiety and depression, are prevalent in AYALHIV, with a pooled prevalence of 26.1% (95% CI 18.9 to 34.8).[2]

However, there is a dearth of integrated programmes combining HIV and mental healthcare.[1 2 5] Importantly, many mental health conditions diagnosed in adults emerge in late adolescence and young adulthood, and effective management at this stage can prevent long-term mental illness.[2 5] Systematic reviews have demonstrated that access to mental healthcare by AYALHIV is associated with positive outcomes across the treatment continuum, including increased treatment initiation, increased adherence to care, viral load control, reduced morbidity and mortality and retention in care.[1 5]

Mental health endpoints within HIV care have been traditionally conceptualised as improvements in negative psychiatric symptomatology.[6 8] For example, success in psychotherapies is invariably benchmarked against declines in anxiety, depression, post-traumatic stress disorders, and other negative psychological indices.[6] However, focusing on negative indices misses the opportunity to capture the multidimensionality of mental health. A holistic mental health evaluation requires a comprehensive focus on both negative and positive mental health constructs,[6 8] and recognition of this has resulted in a shift towards positive psychology, a framework that emphasises increasing human well-being and positive functioning.[6] Positive mental health interventions (PMHIs) are anchored on the need to improve and evaluate human strengths and capabilities as enshrined in positive outcomes such as self-esteem, resilience, hope, self-worth, social resources and flourishing.[6 8 9] For instance, studies have shown that people living with a chronic condition (eg, HIV) develop resilience with time.[6 9] Resilience is defined as the '…positive psychological, behavioural, and/or social adaptation in the face of stressors and adversities…', and is a known buffer to stressful life events.[9] The resilience developed in navigating the challenges of living with a chronic condition is potentially transferable into everyday functioning.[9] Positive psychology interventions (eg, resilience-building approaches) are central to prevention health promotion and act as an entry point to stepped-up care for mental problems in routine HIV care.[6]

With the proliferation of PMHIs comes the need to routinely evaluate the clinical endpoints from both the clients' and therapists' perspectives.[10] The patient-reported outcomes revolution is hinged on assessing treatment success or lack thereof from the patient perspective.[8 10] Patient-reported outcomes facilitate patient–clinician communication and clinical decision-making; this is pivotal to patient-centred care.[8] The patient's evaluation of their health, treatment expectations and outcomes are contingent on the availability of validated and reliable outcome measures.[8] The last few decades have seen a proliferation of positive psychology outcome measures.[11] However, there is limited understanding of the salient positive psychological constructs linked to AYALHIV's improved well-being and health-related quality of life. Wayant *et al* (2021), in their scoping review, mapped 15 positive psychological constructs

associated with increased quality of life and survival in AYAL with cancer.[12] Well-being, personal growth, hope, meaning in life, self-esteem, vitality and optimism were the most cited positive constructs;[12] these constructs are potentially transferable to AYALHIV. Conversely, etiological differences between cancer and HIV could also lead to differences in lived experiences, resulting in differential perceptions in positive psychological constructs.[12] For instance, HIV-related stigma (often associated with issues related to HIV's potential infectiousness) may have a greater impact on mental health functioning in AYALHIV[13 14] when compared with the effects of cancer-related stigma.[15] It is thus critical to contextualise the impacts of positive psychological outcomes in AYALHIV. Govindasamy *et al* (2021) performed a mixed-methods systematic review to explore correlates of well-being among AYALHIV in SSA to inform econometric evaluations.[13] The review showed that social support, belonging, purpose in life and self-acceptance optimise well-being in AYALHIV.[13] Although the review provides essential insights, the sole focus on well-being limits our comprehensive understanding of the spectrum of positive psychological constructs in AYALHIV residing in SSA. There is a need to build on Govindasamy *et al.*'s work to understand positive psychological constructs in HIV care for AYALHIV aged 10–24 years, holistically. Also, there is a paucity of collective evidence of the psychometric robustness of the positive psychological outcomes used in AYALHIV. Some of the available generic outcomes may not comprehensively reflect the nuances of living with HIV.[13] This mixed review, therefore, seeks to:

1. Identify positive psychological outcomes and corresponding outcome measures used in AYALHIV in SSA.
2. Critically appraise the psychometric properties of the identified positive psychology outcomes used in AYALHIV.
3. Map factors associated with the identified positive psychological constructs.
4. Glean an item bank from outcomes with robust psychometrics.

The proposed review will map positive psychological constructs in AYALHIV in SSA by identifying commonly applied constructs and developing an evidence map/conceptual framework for measuring positive psychological outcomes. Furthermore, the review will strengthen the measurement of positive mental health constructs to improve the well-being of AYALHIV. The review will also assist in identifying psychometrics that may require further evaluation and systematically identify factors influencing positive psychological function in AYALHIV, a group with disproportionately poor HIV treatment outcomes.[4 13]

More importantly, the initial item bank will inform the development of a context-specific positive psychology outcome measure for routine clinical and research use. The resultant outcome measure will contribute to the evidence base of the utility of positive psychological interventions in AYALHIV residing in low-income settings.

**Table 1** CINAHL search strategy

| Search # | Query |
|---|---|
| 1 | adolescen* OR juvenile* OR teen* OR youth* OR 'young person*' OR 'young people' OR Adolescence OR Young Adult |
| 2 | Hiv OR hiv-1* OR hiv-2*OR hiv1 OR hiv2 OR 'HIV infect*' OR 'human immunodeficiency virus' OR 'human immunodeficiency virus' OR 'human immuno-deficiency virus' OR 'human immune-deficiency virus' OR 'acquired immunodeficiency syndrome' OR 'acquired immunodeficiency syndrome' OR 'acquired immuno-deficiency syndrome' OR 'acquired immune-deficiency syndrome' OR HIV Infections OR Human Immunodeficiency Virus |
| 3 | 'human immun*' AND 'deficiency virus' |
| 4 | hiv or aids or acquired human immunodeficiency syndrome or human immunodeficiency virus OR HIV/AIDS |
| 5 | 'acquired immun*' AND 'deficiency syndrome' |
| 6 | 2 OR 3 OR 4 0R 5 |
| 7 | hope* OR optimis* OR resilien* OR cope OR coping OR gratitude OR grateful OR happiness OR joy OR gladness OR satisf* OR 'self efficacy' OR self-efficacy OR content*OR wellbeing OR well-being OR 'self acceptance' OR self-acceptance OR 'self esteem' OR self-esteem OR 'self concept' OR self-concept OR 'self confidence' OR self-confidence OR 'self perception' OR self-perception OR 'self worth' OR self-worth OR 'personal growth' OR tranquil* OR perseverance OR vitality OR meaning OR 'social* inclus*' OR 'social participation' OR 'social engagement' OR 'social support' OR 'self care' OR self-care OR 'positive attitude' OR 'positive thinking' OR 'positive mindset' OR mindfulness OR empower*OR love OR spiritual* OR 'community integration' OR 'community participation' OR humour OR dignity OR pleasure OR creativ* OR transcend* OR goal* OR Psychological Well-Being OR Hope OR Optimism OR Coping OR Human Dignity OR Hardiness OR Adaptation, Psychological OR Happiness OR Personal Satisfaction OR Self Efficacy OR Self Concept OR Social Inclusion OR Social Participation OR Self Care OR Mindfulness OR Empowerment OR Love OR Spirituality OR Social Networks OR Pleasure OR Creativeness OR Self Transcendence OR (Goals and Objectives) |
| 8 | hope* OR optimis* OR optimism OR resilience OR resilien* OR resilient OR cope OR coping OR gratitude OR grateful OR happiness OR joy OR gladness OR (life satisfaction) OR satisfaction OR satisf* OR 'self efficacy' OR self-efficacy OR content OR contentment OR content*OR wellbeing OR well-being OR 'self acceptance' OR self-acceptance OR 'self esteem' OR self-esteem OR 'self concept' OR self-concept OR 'self confidence' OR self-confidence OR 'self perception' OR self-perception OR 'self worth' OR self-worth OR 'personal growth' OR tranquillity OR tranquil* OR perseverance OR vitality OR meaning OR 'social* inclus*' OR 'social participation' OR 'social engagement' OR 'social support' OR 'self care' OR self-care OR 'positive attitude' OR 'positive thinking' OR 'positive mindset' OR mindfulness OR empowerment OR empower*OR love OR spirituality OR spiritual* OR 'community integration' OR 'community participation' OR humour OR dignity OR pleasure OR creativ* OR transcend* OR goal* OR Psychological Well-Being OR Hope OR Optimism OR Coping OR Human Dignity OR Hardiness OR Adaptation, Psychological OR Happiness OR Personal Satisfaction OR Self Efficacy OR Self Concept OR Social Inclusion OR Social Participation OR Self Care OR Mindfulness OR Empowerment OR Love OR Spirituality OR Social Networks OR Pleasure OR Creativeness OR Self Transcendence OR (Goals and Objectives) OR growth mindset OR positive mindset OR (*mindset) OR flourishing OR flourish* OR thriving OR thriv* |
| 9 | 7 OR 8 |
| 10 | Angola or Benin or Botswana or Burkina Faso or Burundi or Cameroon or Cape Verde or Central African Republic or CHAD or Comoros or Congo or Congo Democratic Republic or Djibouti or Equatorial Guinea or Eritrea or Ethiopia or Gabon or Gambia or Ghana or Guinea or Guinea-Bissau or Cote d'Ivoire or Ivory Coast or Kenya or Lesotho or Liberia or Madagascar or Malawi or Mali or Mozambique or Namibia or Niger or Nigeria or Sao tome and Principe or Rwanda or Senegal or Seychelles or Sierra Leone or Somalia or South Africa or South Sudan or Sudan or Swaziland or Tanzania or Togo or Uganda or Zambia or Zimbabwe |
| 11 | 'Africa, South of the Sahara' OR 'sub-Saharan Africa' |
| 12 | 10 OR 11 |

*Adapted from Terwee *et al*.[19]

## METHODS

### Overview

This mixed review will be done in two sequential and complementary phases. First, a scoping review will identify positive psychological outcomes used in AYALHIV in SSA and map the constructs onto the corresponding measures. The scoping review will be performed per Preferred Reporting Items for Systematic reviews and Meta-Analyses extension for Scoping Reviews (PRISMA-ScR) guidelines—see online supplemental file 1.[16] The second phase will systematically evaluate the psychometric properties of the outcomes identified from the scoping review. The evaluation of psychometrics of the outcome measures will be performed and reported according to the Preferred Reporting Items of Systematic Reviews and Meta-Analyses Protocol (PRISMA-P) guidelines—see online supplemental file 2.[17] Where appropriate, we will outline specific methodological considerations unique to each phase.

### Eligibility criteria

The following criteria will be applied in selecting articles.

### Study designs/interventions

For the scoping review, we will include all quantitative designs, mixed-methods, qualitative studies exploring the positive psychological phenomenon in AYALHIV in SSA, and grey literature (eg, blogs and websites). For the systematic review, only quantitative designs will be included. Systematic reviews, editorials, case studies and study protocols will be excluded from the scoping and systematic reviews.

### Participants/settings

For both phases of the mixed review, we will analyse all studies reporting on using and evaluating positive psychological constructs in AYALHIV (10–24 years old) in SSA across all settings. We focus on AYALHIV as it is the group with the greatest burden of HIV globally.[4 13] We anticipate that some studies will contain data on AYALHIV and other age bands (eg, children, middle-aged adults). In such cases, an article will be considered for review if the average age is within the 10–24 years range or if over 50% of the participants are AYALHIV.

**Table 2** Operational definitions of psychometric properties[19 20]

| Term | | | |
|---|---|---|---|
| **Domain** | **Measurement property** | **Aspect of a measurement property** | **Definition** |
| Reliability | | | The degree to which the measurement is free from measurement error |
| Reliability (extended definition) | | | The extent to which scores for patients who have not changed are the same for repeated measurement under several conditions: for example, using different sets of items from the same patient-reported outcome measure (PROM) (internal consistency); over time (test–retest); by different persons on the same occasion (inter-rater) or by the same persons (ie, raters or responders) on different occasions (intrarater) |
| | Internal consistency | | The degree of the inter-relatedness among the items |
| | Reliability | | The proportion of the total variance in the measurements which is due to 'true'* differences between patients |
| | Measurement error | | The systematic and random error of a patient's score that is not attributed to true changes in the construct to be measured |
| Validity | | | The degree to which a PROM measures the construct(s) it purports to measure |
| | Content validity | | The degree to which the content of a PROM is an adequate reflection of the construct to be measured |
| | | Face validity | The degree to which (the items of) a PROM indeed looks as though they are an adequate reflection of the construct to be measured |
| | Construct validity | | The degree to which the scores of a PROM are consistent with hypotheses *(for instance with regard to internal relationships, relationships to scores of other instruments or differences between relevant groups)* based on the assumption that the PROM validly measures the construct to be measured |
| | | Structural validity | The degree to which the scores of a PROM are an adequate reflection of the dimensionality of the construct to be measured |
| | | Hypotheses testing | Item construct validity |
| | | Cross-cultural validity | The degree to which the performance of the items on a translated or culturally adapted PROM are an adequate reflection of the performance of the items of the original version of the PROM |
| | Criterion validity | | The degree to which the scores of a PROM are an adequate reflection of a 'gold standard' |
| Responsiveness | | | The ability of a PROM to detect change over time in the construct to be measured |
| | Responsiveness | | Item responsiveness |
| Interpretability† | | | Interpretability is the degree to which one can assign qualitative meaning—that is, clinical or commonly understood connotations—to a PROM's quantitative scores or change in scores |

*The word 'true' must be seen in the context of the CTT, which states that any observation is composed of two components—a true score and error associated with the observation. 'True' is the average score that would be obtained if the scale were given an infinite number of times. It refers only to the consistency of the score, and not to its accuracy.
†Interpretability is not considered a measurement property, but an important characteristic of a measurement instrument
CTT, classical test theory; PROM, patient-reported outcome measure.

**Table 3** Updated criteria for good measurement properties[21 22]

| Measurement property | Rating | Criteria |
|---|---|---|
| Structural validity | + | CTT:<br>CFA: CFI or TLI or comparable measure>0.95 OR RMSEA <0.06 OR SRMR<0.08<br>IRT/Rasch:<br>No violation of unidimensionality: CFI or TLI or comparable measure>0.95 OR RMSEA<0.06 OR SRMR<0.08<br>*AND* no violation of *local independence*: residual correlations among the items after controlling for the dominant factor<0.20 OR Q3's<0.37<br>*AND* no violation of *monotonicity*: adequate looking graphs OR item scalability>0.30<br>*AND* adequate *model fit*:<br>IRT: $\chi^2$>0.01<br>Rasch: infit and outfit mean squares≥0.5 and ≤ 1.5 OR Z-standardised values > −2 and<2 |
| | ? | CTT: not all information for '+' reported IRT/Rasch: model fit not reported |
| | − | Criteria for '+' not met |
| Internal consistency | + | At least low evidence for sufficient structural validity AND Cronbach's alpha(s) ≥ 0.70 for each unidimensional scale or subscale 6 |
| | ? | Criteria for 'at least low evidence for sufficient structural validity' not met |
| | − | At least low evidence for sufficient structural validity AND Cronbach's alpha(s) <0.70 for each unidimensional scale or subscale |
| Reliability | + | ICC or weighted Kappa≥0.70 |
| | ? | ICC or weighted Kappa not reported |
| | − | ICC or weighted Kappa<0.70 |
| Measurement error | + | SDC or LoA<MIC |
| | ? | MIC not defined SDC or LoA>MIC |
| | − | |
| Hypotheses testing for construct validity | + | The result is in accordance with the hypothesis |
| | ? | No hypothesis defined (by the review team) |
| | − | The result is not in accordance with the hypothesis |
| Cross-cultural validity/ measurement invariance | + | No important differences found between group factors (such as age, gender and language) in multiple group factor analysis OR no important DIF for group factors (McFadden's $R^2$<0.02) |
| | ? | No multiple group factor analysis OR DIF analysis performed |
| | − | Important differences between group factors OR DIF were found |
| Criterion validity | + | Correlation with gold standard≥0.70 OR AUC≥0.70 |
| | ? | Not all information for '+' reported |
| | − | Correlation with gold standard<0.70 OR AUC<0.70 |
| Responsiveness | + | The result is in accordance with the hypothesis OR AUC≥0.70 |
| | ? | No hypothesis defined (by the review team) |
| | − | The result is not in accordance with the hypothesis OR AUC<0.70 |

ACU, area under the curve; CFA, confirmatory factor analysis; CFI, Comparative Fit Index; CTT, classical test theory; DIF, differential item functioning; ICC, intraclass correlation coefficient; IRT, item response theory; LoA, limits of agreement; MIC, minimal important change; RMSEA, root mean square error of approximation; SDC, smallest detectable change; SRMR, standardised root mean residuals; TLI, Tucker-Lewis Index.

## Language

We will restrict the analysis to articles published in English for both phases of the mixed review. We do not have the resources to analyse articles published in other languages.

## Information sources

Peer-reviewed articles will be searched/retrieved from these electronic databases: PubMed, Scopus, Web of Science, Africa-Wide Information, CINAHL, PsychINFO and Google Scholar. Databases will be searched from inception through 30 September 2022. Where only an abstract is available online, and information regarding psychometrics is neither clear nor available from the text, an attempt to contact the lead author will be made requesting the full article to ensure literature saturation and a truthful rating. The article will be excluded from the review if there is no response in 2 weeks following three email reminders. We will also review grey literature using the Google Scholar search engine to search potential databases such as university databases, research reports, preprints, newsletters and bulletins, policy briefs, guidelines and conference proceedings for articles. For completeness, we will also perform both backwards and forward searches of the reference lists of identified articles and databases, respectively. Finally, we will also contact experts implementing PMHIs to check for articles we may have missed using the proposed search strategy.

## Search strategy

For the scoping review, as an illustration, articles in CINAHL will be searched using the AND Boolean logic operators, that is, 1 AND 6 AND 9 AND 12 (table 1). The search strategy will be amended for the systematic review

**Table 4** GRADE checklist—best evidence synthesis[23]

| Quality level | Definition/criterion |
| --- | --- |
| High | We are very confident that the true measurement property lies close to that of the estimate of the measurement property |
| Moderate | We are moderately confident in the measurement property estimate: the true measurement property is likely to be close to the estimate of the measurement property, but there is a possibility that it is substantially different |
| Low | Our confidence in the measurement property estimate is limited: the true measurement property may be substantially different from the estimate of the measurement property |
| Very low | We have very little confidence in the measurement property estimate: the true measurement property is likely to be substantially different from the estimate of the measurement property |

component to include additional constructs identified through the scoping review.

### Data management

Retrieved articles will be imported into the Mendeley reference manager, which is password-protected. The articles will also be synchronised onto Mendeley and Dropbox cloud storage platforms and backed-up onto a password-encrypted external hard drive. All collaborators will have full access/administrative privileges to the shared Dropbox folder for the present systematic review. A trail/history of the electronic searches will also be saved on users' PubMed, Scopus and EBSCOhost accounts. We will also print summaries of all the searches to enhance the data capturing of the search records.

### Data collection process

The data collection process will be conducted in three stages, that is, article retrieval, screening and data extraction. These processes will invariably be similar for the scoping and systematic review phases. Here, we describe these processes and highlight, where appropriate, differences in the two phases of the review.

### Article retrieving

Two researchers (SS and VS) will independently search articles using a predefined search strategy. The lead author (JD) will then import the searches into Mendeley and remove duplicates.

### Screening

On completion of article retrieving, another set of independent researchers (SB and WM) will screen the articles by title and abstract using Rayyan software.[18] To increase methodological rigour, both researchers will independently review all retrieved articles, including documenting reasons for exclusion. Rayyan software automatically collates the number of hits assigned different ratings by the reviewers. Discrepancies will be resolved through discussion, and where consensus is not reached, a more senior researcher (WM) will make the final decision. JD and SS will then perform backwards and forward citation searches to identify other potential articles. Two

senior researchers (FM and WM) will review the list of identified articles afterwards to check for the completeness of the searches.

### Data extraction

Once searches are finalised, two researchers (FM and NW) will retrieve the full articles and independently extract data from articles meeting the inclusion criteria. Data extraction will be performed in duplicate. Disagreements during data extraction will be resolved through consensus, and more senior researchers (FM and WM) will make the final decisions if any impasses occur. For both phases of the review, we will extract the following information per study: research setting and design, study sample and participants' demographics. For the scoping review, we will additionally extract information on the conceptualisations/definitions and factors associated with positive psychological constructs. For the systematic review component, we will extract information on the mode of administration, the number of items, descriptions of domains, scoring and interpretation of scores and whether measures are free to use or require a license fee or other payment.

### Charting/outcomes and prioritisation

The conceptualisation of positive psychological constructs and the psychometric properties of the identified outcomes will be the primary outcome measures for the scoping and systematic review phases, respectively. For the systematic review, the clinical utility of the identified outcome measures will be the secondary outcome. See table 2 for operational definitions of psychometric properties for the systematic review component.[19 20]

### Risk of bias—individual studies

The scoping review aims to understand the conceptualisation of AYALHIV's positive psychological constructs. Consequently, we will not perform any risk of bias (RoB) assessments. However, the systematic review component aims to synthesise the evidence of psychometric robustness necessitating RoB assessment. We will use the revised COnsensus-based Standards for the selection of health Measurement INstruments (COSMIN) checklist to assess the RoB across studies retrieved for psychometric evaluation.[19 20] The COSMIN methodology consists of three steps. The checklist consists of methodological benchmarks for 10 psychometric properties, which are categorised into three major groups, that is, content validity (eg, patient-reported outcome measure development), internal structure (eg, structural validity) and other psychometrical properties (eg, criterion validity).[19 20] Each psychometric property is rated using a preset criterion, and using the principle of 'worse score counts', the lowest rating is ascribed as the overall methodological quality rating.[19] Methodological quality is rated on a four-point Likert scale, that is, 'inadequate', 'doubtful', 'adequate' and 'very good'; the higher the rating, the lower the RoB.[19 20] We anticipate that not all details may be recorded for the

retrieved articles, especially for studies whose primary aim was not psychometric evaluation. We will, therefore, contact the corresponding author to achieve the most truthful rating of the psychometric property to decrease bias during analysis.

## Quality of psychometric properties and data extraction

The quality of psychometrical properties will be evaluated using an updated, hybrid checklist based on previous work by Terwee et al[21] and Prinsen et al[22] (see table 3). Each psychometric property will be rated as sufficient (+), insufficient (−) or indeterminate (?).[20] Positive ratings represent high-quality psychometrics.[20]

## Best evidence synthesis

Initially, we will develop a conceptual framework for synthesising both qualitative and quantitative studies retrieved. We will then apply a narrative synthesis to map the key 'themes/constructs' emerging from the scoping review. The conceptual framework/model will map the constructs and subsequently guide the psychometric evaluation. The collective evidence per psychometric property per outcome will be synthesised using the modified Grading of Recommendations Assessment, Development and Evaluation (GRADE) checklist,[23] as outlined in table 4. The modified GRADE will then be used to collate the RoB results and the quality of psychometric ratings to qualitatively synthesise/summarise the quality of evidence per psychometric property across studies. The quality of evidence per psychometrical property will be classified as very low, low, moderate or high.[23]

## Patient and public involvement statement

We will work collaboratively with AYALHIV during data collection and dissemination. AYALHIV representatives previously trained and involved in systematic reviews will assist with article screening. We will cocreate the dissemination plans; for instance, adolescents and young adults with lived experiences will be involved in codeveloping output animation and contributing to the project blogs, among other dissemination activities.

## Ethics and dissemination

No ethical approvals are needed as this is a secondary study. The proposed mixed review will map and appraise the collective evidence of the psychometric robustness of positive psychological outcomes used in AYALHIV. The proposed review builds on recommendations of systematic reviews on the need to measure positive psychological constructs across diverse populations objectively. This is important given the need to use valid and reliable outcomes in understanding the positive effects of living with HIV. The review will also assist in identifying psychometrically robust outcomes to inform an item bank to adapt a context-specific outcome measure for AYALHIV in low-resource settings. For example, we will consolidate all self-esteem outcome measures and categorise items from multiple outcomes into common factors/'themes'. Also, the review will systematically identify factors influencing well-being in AYALHIV. The outputs will collectively inform the development, implementation and evaluation of bespoke PMHIs for AYALHIV, hence a need for a multimodal dissemination plan to reach multiple stakeholders. This review is attached to ongoing work in which AYALHIV are collaboratively engaged. It is part of a larger study to explore various constructs to understand how they improve AYALHIV's health outcomes. We have already recruited AYALHIV to serve as a Youth Expert Panel (YEP). The YEP functions as both a guide to the study/research process and an additional group of analysts and discussants to examine the emerging analysis and findings. We will publish the outcomes in a peer-reviewed journal. Additionally, we will disseminate the outcomes through social media, policy briefs and blogs.

**Author affiliations**
¹Primary Healthcare Sciences, University of Zimbabwe Faculty of Medicine, Harare, Zimbabwe
²Centre for Sexual Health and HIV/AIDS Research, Harare, Zimbabwe
³International Public Health, Liverpool School of Tropical Medicine, Liverpool, UK
⁴Centre for Intelligent Healthcare, Coventry University, Coventry, UK
⁵MRC International Statistics and Epidemiology Group, London School of Hygiene & Tropical Medicine, London, UK
⁶ZVANDIRI, Harare, Zimbabwe
⁷Global Health and Development, London School of Hygiene and Tropical Medicine, London, UK
⁸School of Public Health, University of Sydney, Sydney, New South Wales, Australia

**Acknowledgements** We would like to acknowledge Alison Derbyshire, an Academic Liaison & Training Specialist at the Liverpool School of Tropical Medicine Library, for assisting with refining the search strategies.

**Collaborators** N/A.

**Contributors** JMD was primarily responsible for protocol writing. JMD, FMC, FM, SS, VC, NW, SB, and WM were involved in the conceptualisation of the study and editing all protocol manuscript versions. JMD will search the literature and data management. FMC, FM, SS, VC, NW, SB, and WM will be responsible for article screening and quality assurance, data extraction and qualitative synthesis.

**Funding** This project was made possible through the support of a grant from Templeton World Charity Foundation, Inc (funder DOI 501100011730) through grant https://doi.org/10.54224/20629. The opinions expressed in this publication are those of the authors and do not necessarily reflect the views of Templeton World Charity Foundation, Inc. For the purpose of open access, the authors have applied a CC-BY public copyright license to any author accepted manuscript version arising from this submission.

for any error and/or omissions arising from translation and adaptation or otherwise.

**ORCID iDs**
Jermaine M Dambi http://orcid.org/0000-0002-2446-7903
Frances M Cowan http://orcid.org/0000-0003-3087-4422
Faith Martin http://orcid.org/0000-0002-0141-1210
Sharon Sibanda http://orcid.org/0000-0002-6595-4097
Victoria Simms http://orcid.org/0000-0002-4897-458X
Nicola Willis http://orcid.org/0000-0003-0452-0196
Sarah Bernays http://orcid.org/0000-0001-7628-8408
Webster Mavhu http://orcid.org/0000-0003-1881-4398

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
