## [Reviewer comments · BMJ Open]

ARTICLE DETAILS

TITLE (PROVISIONAL)	A conceptualisation and psychometric evaluation of positive psychological outcome measures used in adolescents and young adults living with HIV: a mixed scoping and systematic review protocol.
AUTHORS	Dambi, Jermaine; Cowan, F; Martin, Faith; Sibanda, Sharon; Simms, Victoria; Willis, Nicola; Bernays, Sarah; Mavhu, Webster

VERSION 1 – REVIEW

REVIEWER	Ishrat Yousaf Capital University of Science and Technology
REVIEW RETURNED	04-Aug-2022

GENERAL COMMENTS	I have reviewed this article. From the perspective of Mental health research in HIV/AIDS, this problem is an appropriate, meaningful and has applied significance specifically in Low and Middle income countries. This is an interesting study and the authors have collected a unique dataset using mix method approach. The paper is generally well written and structured. However, in my opinion the paper has some shortcomings in regards to some data analyses and text, however, I feel this unique study will be utilized to its full extent.
---

REVIEWER	Kaitlyn Atkins Johns Hopkins Bloomberg School of Public Health, Department of Epidemiology
REVIEW RETURNED	07-Aug-2022

GENERAL COMMENTS	Manuscript ID: bmjopen-2022-066129 This review protocol responds to an emerging literature on positive psychosocial outcomes among AYA living with HIV. The planned review is multiphase and includes (1) a scoping review to identify positive psychological constructs used among AYALHIV and (2) a systematic review to evaluate the psychometric properties of these constructs as applied in the literature. The protocol is sound and the review is timely. Below are suggestions and clarifications that could improve the protocol. All of these comments pertain to the Methods section. 1. The methods seem to largely focus on the procedures for the systematic review (and not the scoping review). The scoping review is mentioned at the end under “Best evidence thesis”—perhaps a separate section briefly delineating scoping review methods should be placed at the beginning of Methods? Currently, it is a bit challenging to understand how the scoping review output informs the systematic review, or if these are rather complementary but separate reviews.2. The authors state they “will map the positive psychological
---

	constructs and identify the corresponding outcome measures.” What do the authors mean by “map the constructs?” Does this mean “map the constructs onto corresponding outcomes?” Some rephrasing would help clarify – this also applies to similar phrases in the abstract and introduction. 3. Eligibility criteria: Broadly speaking, I think it would be useful to report these separately by study phase. a. Why are qualitative studies being included in the second phase of the review, which focuses on psychometrics? This could be clarified if study design inclusion criteria were reported for each review specifically. b. Some more specificity around use of grey literature would be helpful. Specifically what types of grey literature will be eligible for inclusion (i.e., published reports vs blogs or Web pages)? 4. Search strategy: a. Did the authors work with a reference librarian to develop these search terms? b. Is there a plan to revise the search strategy after the scoping review? It may be unlikely, but is it possible that the scoping review identifies other constructs not included in the current search string? c. It is likely impossible to include every possible positive psychology construct in the search string. While it may be too late to revise the search at this stage, some concepts that could be missing are growth mindset, flourishing, thriving 5. Screening: I suggest this be its own section with a few more details added. a. Will title and abstract screening be done in duplicate or will the two authors each screen half of the retrieved articles? I think either is acceptable at this stage but it is not clear. b. Do you plan to include a second-stage screening (i.e. full text screening) before extraction? As written, it seems that the only screening stage is title/abstract screening, followed by extraction from full-text articles. Based on the breadth of this search an additional layer of screening could help reduce burden at the extraction stage and would improve rigor. Typically this would be done in duplicate with a pre-specified process for achieving consensus if not reached. 6. Data extraction: The inclusion of cross-checks from senior authors is a great step to improve rigor. I see two reviewers will do extraction; does this mean extraction will be done in duplicate with consensus achieved through discussion, or will each reviewer separately extract data from half of the included articles? This is not currently clear. a. What is the plan for contacting authors to retrieve information about psychometric properties not reported in-text (i.e., time allotted to respond, plan for categorizing non-responders)? 7. Included discussion of co-creation with AYALHIV is fantastic but a bit unclear. Are there specific ways this involvement will take shape? What relationships does this team have with AYALHIV, or how does the team plan to form those if not established? For example, will AYALHIV be invited to join an advisory group to help with the review, or is this review attached to ongoing work in which AYALHIV are collaboratively engaged? Minor comments:  • At times the authors use “young people living with HIV” and at other times they use “AYALHIV.” For consistency and to reduce word count, suggest using AYALHIV throughout the manuscript and clearly defining who is included in this group.
--	--

	 • In Table 2, I am not sure both rows are necessary for “responsiveness”—the second seems duplicative. • Table 3 includes several superscripted references but these do not appear to correspond to references cited or footnotes in the table. Please double-check • The authors mention plans to develop an item bank; any relevant steps to develop this could be included in the protocol if not already covered by what is written.
--	--

VERSION 1 – AUTHOR RESPONSE

REVIEWER 1

I have reviewed this article. From the perspective of Mental health research in HIV/AIDS, this problem is an appropriate, meaningful and has applied significance specifically in Low and Middle income countries. This is an interesting study and the authors have collected a unique dataset using mix method approach. The paper is generally well written and structured. However, in my opinion the paper has some shortcomings in regards to some data analyses and text, however, I feel this unique study will be utilised to its full extent.

Response: Thank you, we have edited the data analyses section to reflect the scoping and systematic review components of the mixed review.

REVIEWER 2

This review protocol responds to an emerging literature on positive psychosocial outcomes among AYA living with HIV. The planned review is multiphase and includes (1) a scoping review to identify positive psychological constructs used among AYALHIV and (2) a systematic review to evaluate the psychometric properties of these constructs as applied in the literature.

The protocol is sound, and the review is timely. I have attached some suggestions and clarifications that could improve the protocol. Thank you for the opportunity to review!

Response: Thank you.

1. The methods seem to largely focus on the procedures for the systematic review (and not the scoping review). The scoping review is mentioned at the end under “Best evidence thesis”—perhaps a separate section briefly delineating scoping review methods should be placed at the beginning of Methods? Currently, it is a bit challenging to understand how the scoping review output informs the systematic review, or if these are rather complementary but separate reviews.

Response: Thank you for noting the gap; we have updated the methods section to distinguish the two phases of the mixed review. There are instances where there is an overlap of methods. We have edited the methods section to be specific where there are differences. For instance, under the risk of bias assessment section, we have added the following statement, “The scoping reviews aim to understand the conceptualisation of positive psychological constructs by YLHIV in SAA. Consequently, we will not perform any risk of bias (RoB) assessments. However, the systematic review component aims to synthesise the evidence of psychometric robustness necessitating RoB assessment. We will use the revised CONsensus-based Standards for the selection of health Measurement INSTRUMENTS (COSMIN) checklist to assess the RoB across studies retrieved for psychometric evaluation”. (page 10: lines 247–252). Additionally, we have edited some of the sections to reflect the dual nature of the review; for instance, the outcomes and prioritisation section has been changed to Charting/outcomes and prioritisation; see page 9: line 232.

2. The authors state they “will map the positive psychological constructs and identify the corresponding outcome measures.” What do the authors mean by “map the constructs?” Does this mean “map the constructs onto corresponding outcomes?” Some rephrasing would help clarify – this also applies to similar phrases in the abstract and introduction.

Response: Thank you for pointing out the ambiguity. As suggested, we have rephrased the statement to improve clarity. It now reads “...identify positive psychological outcomes used in AYALHIV in sub-Saharan Africa and map the constructs onto corresponding measures”. page 6: lines 149-150.

3. Eligibility criteria: Broadly speaking, I think it would be useful to report these separately by study phase.

Response: Thank you for the suggestion; indeed, separating the descriptions of the eligibility criteria by phase will improve clarity. Also, there are instances where the eligibility criteria are uniform across the phases, for example, eligibility by language. To improve clarity, we have added the following text to distinguish differences in criteria between the two phases:

- For the scoping review, we will include all quantitative designs, mixed methods, and qualitative studies exploring the positive psychological phenomenon in AYALHIV in SSA, and grey literature (e.g. blogs, websites). For the systematic review, only quantitative designs will be included. Systematic reviews, editorials, case studies and study protocols will be excluded from the scoping and systematic reviews. page 6: lines 163–166.
 - For participants/settings, we have added the following text: “For both phases of the mixed review, we will analyse all studies reporting on using and evaluating positive psychological constructs in AYALHIV in SSA across all settings”. page 6: lines 168–169.
 - For language, we have added the following text , “We will restrict the analysis to articles published in English **for both phases of the mixed review**”. (page 6: line 175).
- a. Why are qualitative studies being included in the second phase of the review, which focuses on psychometrics? This could be clarified if study design inclusion criteria were reported for each review specifically.

Response: Thank you for noting the error; qualitative studies will only be considered for the scoping review. To improve clarity, we have re-written the following text in the Study designs/interventions section, “For the scoping review, we will include all quantitative designs, mixed methods, qualitative studies exploring the positive psychological phenomenon in AYALHIV in SSA, and grey literature (e.g. blogs, websites). For the systematic review, only quantitative designs will be included”. page 6: lines 163–166.

- b. Some more specificity around use of grey literature would be helpful. Specifically what types of grey literature will be eligible for inclusion (i.e., published reports vs blogs or Web pages)?

Response: Thank you, we have edited the statement to, “We will also review grey literature using the Google Scholar search engine to search potential databases such as university databases, research reports, pre-prints, newsletters and bulletins, policy briefs, guidelines and conference proceedings for articles”. page 7: lines 184–186.

4. Search strategy:

- a. Did the authors work with a reference librarian to develop these search terms?

Response: Yes, the search strategy was finalised in consultation with a librarian. We have amended our acknowledgements section to reflect the support provided. The acknowledgement section now reads, “We would like to acknowledge Alison Derbyshire, an Academic Liaison & Training Specialist at the Liverpool School of Tropical Medicine Library, for assisting with refining the search strategies”. page 14: lines 342–344.

- b. Is there a plan to revise the search strategy after the scoping review? It may be unlikely, but is it possible that the scoping review identifies other constructs not included in the current search string?

Response: Thank you for the suggestion; we will amend the search strategy should we identify additional constructs. We have added the following text in the search strategy section, “The search strategy will be amended for the systematic review component to include additional constructs identified through the scoping review”. page 7: lines 192–193.

- c. It is likely impossible to include every possible positive psychology construct in the search string. While it may be too late to revise the search at this stage, some concepts that could be missing are growth mindset, flourishing, thriving.

Response: Indeed, we concur that it may not be possible to include all positive constructs, as the field of positive psychology is still evolving, particularly in the sub-Saharan Africa region, our study focus. We have added the suggested outcomes as we want to capture as many constructs as possible. We have added the following to the search strategy, "OR growth mindset OR positive mindset OR (*mindset) OR flourishing OR flourish* OR thriving OR thrive*" page 7-8, lines 195–196.

5. Screening: I suggest this be its own section with a few more details added.

a. Will title and abstract screening be done in duplicate or will the two authors each screen half of the retrieved articles? I think either is acceptable at this stage but it is not clear.

Response: Thank you for noting the ambiguity. We have added the following text, to increase clarity; "To increase methodological rigour, both researchers will independently review all retrieved articles, including documenting reasons for exclusion. Rayyan software automatically collates the number of hits with indifferent ratings. Discrepancies will be resolved through discussion, and where consensus is not reached, a more senior researcher (WM) will make the final decision". page 8-9: lines 214–218.

b. Do you plan to include a second-stage screening (i.e. full text screening) before extraction? As written, it seems that the only screening stage is title/abstract screening, followed by extraction from full-text articles. Based on the breadth of this search an additional layer of screening could help reduce burden at the extraction stage and would improve rigor. Typically, this would be done in duplicate with a pre-specified process for achieving consensus if not reached.

Response: Thank you for the suggestion. We concur that second-level screening reduces data extraction burden. Our preliminary searches yielded manageable volumes of literature that would not present a burden at the extraction stage, hence the decision to screen by title and abstract.

6. Data extraction: The inclusion of cross-checks from senior authors is a great step to improve rigor. I see two reviewers will do extraction; does this mean extraction will be 2 done in duplicate with consensus achieved through discussion, or will each reviewer separately extract data from half of the included articles? This is not currently clear.

Response: Thank you for noting the ambiguity. We have edited the statement to, "Once searches are finalised, two researchers (FM and NW) will retrieve the full articles and independently extract data from articles meeting the inclusion criteria. Data extraction will be performed in duplicate. Disagreements during data extraction will be resolved through consensus, and more senior researchers (FMC & WM) will make the final decisions if any impasses occur". page 9: lines 222–225.

a. What is the plan for contacting authors to retrieve information about psychometric properties not reported in-text (i.e., time allotted to respond, plan for categorising non-responders)?

Response: Thank you for raising this crucial point that we had overlooked. Our definitions of psychometrics are modelled on the COSMIN taxonomy. Based on the reviewer's suggestion, we will contact developers of the retrieved outcomes to glean data on feasibility parameters as suggested. We have edited the statement to, "Where only an abstract is available online, and information regarding psychometrics is neither clear nor available from the text, an attempt to contact the lead author will be made requesting the full article to ensure literature saturation and truthful rating. The article will be excluded from the review if there is no response in two weeks following three email reminders". Pages 6-7: lines 180–184.

7. Included discussion of co-creation with AYALHIV is fantastic but a bit unclear. Are there specific ways this involvement will take shape? What relationships does this team have with AYALHIV, or how does the team plan to form those if not established? For example, will AYALHIV be invited to join an advisory group to help with the review, or is this review attached to ongoing work in which AYALHIV are collaboratively engaged?

Response: This review is attached to ongoing work in which AYALHIV are collaboratively engaged. It is part of a larger study to explore various constructs to understand how they improve AYALHIV's health outcomes. We have already recruited AYALHIV to serve as a Youth Expert Panel (YEP). The YEP functions as both a guide to the study/research process but also as an additional group of

analysts and discussants to examine the emerging analysis and findings. We now state this in the protocol. page 13: lines 308–312.

Minor comments:

- At times the authors use “young people living with HIV” and at other times they use “AYALHIV.” For consistency and to reduce word count, suggest using AYALHIV throughout the manuscript and clearly defining who is included in this group.

Response: Thank you for noting this inconsistency. We now use AYALHIV throughout the manuscript. We have also clarified that these are 10-24 year-olds living with HIV. page 2: lines 36–37 and page 4 line 74.

- In Table 2, I am not sure both rows are necessary for “responsiveness”—the second seems duplicative.

Response: Indeed, the word is repeated; generally, the table is organised so that the first row gives an operational definition of the psychometric property. The second row then gives the specific aspects of a measurement property. For example, reliability has three measurement aspects, whereas responsiveness has one aspect, hence the repetition of the term.

- Table 3 includes several superscripted references but these do not appear to correspond to references cited or footnotes in the table. Please double-check.

Response: Thank you for spotting this. We have deleted the references and the table is now referenced (page 11: line 269).

- The authors mention plans to develop an item bank; any relevant steps to develop this could be included in the protocol if not already covered by what is written.

Response: Thank you, we have added the following statement as suggested, “For example, we will consolidate all self-esteem outcome measures and categorise items from multiple outcomes into common factors/“themes”. Please refer to page 13: lines 303–305. The precise details of the development of the item bank will be specified in greater detail in a follow on paper outlining the development and validation of a composite positive psychological outcome measure for use in AYALHIV in SSA.

VERSION 2 – REVIEW

REVIEWER	Kaitlyn Atkins Johns Hopkins Bloomberg School of Public Health, Department of Epidemiology
REVIEW RETURNED	09-Sep-2022

GENERAL COMMENTS	All of my comments have been sufficiently addressed. The revised manuscript reads well and is very clear. Well done to the authors for these revisions!
---